

# Marine Organic Aerosols at Mace Head: Effects from Phytoplankton and Source Region Variability

Emmanuel Chevassus[1], Kirsten N. Fossum[1], Darius Ceburnis[1], Lu Lei[1], Chunshui Lin [1,2], Wei Xu[1a]; Colin D.O' Dowd[1], Jurgita Ovadnevaite[1]

[1] School of Natural Sciences, Ryan Institute's Centre for Climate and Air Pollution Studies, University of Galway, Galway, Ireland

[2] Institute of Urban Environment, Chinese Academy of Sciences, Xiamen 361021

[a] Now at: State Key Laboratory of Loess and Quaternary Geology (SKLLQG), Center for Excellence in Quaternary Science and Global Change, Institute of Earth Environment, Chinese Academy of Sciences, Xi'an
710061, China

*Correspondence to:* Jurgita Ovadnevaite (jurgita.ovadnevaite@universityofgalway.ie)

## 2. Abstract

Organic aerosols (OA) are recognised as a significant component of particulate matter (PM), yet, their specific composition and sources, especially over remote areas remain elusive due to the overall scarcity of high-
resolution online data. In this study, positive matrix factorisation was performed on organic aerosol mass spectra obtained from high-resolution time-of-flight aerosol mass spectrometer (HR-ToF-AMS) measurements to resolve sources contributing to the coastal PM. The focus was on a summertime period marked by enhanced biological productivity with prevailing pristine maritime conditions. Four OA factors were deconvolved by the source apportionment model. The analysis revealed primary marine organic aerosols (PMOA) as the
predominant submicron OA at Mace Head during summertime, accounting for 42%. This was trailed by more oxidized oxygenated organic aerosols (MO-OOA) at 32%, methanesulphonic acid organic aerosols (MSA-OA) at 17%, and locally emitted peat-derived organic aerosols (Peat-OA) at 9% of the total OA mass. The total mass concentrations of primary organic aerosols and secondary organic aerosols were overall equal and almost exclusively present in the marine boundary layer consistently with previous findings. This study reveals that OA
not only reflects atmospheric chemistry and meteorology – as evidenced by the significant aging of summertime polar air masses over the North Atlantic, driven by ozonolysis under Greenland anticyclonic conditions - but also serve as indicators of marine ecosystems. This is evident from MSA-OA being notably associated with stress enzyme markers and PMOA showing the typical makeup of largely abacterial phytoplankton extracellular metabolic processes. This study also reveals distinctive source regions within the North Atlantic Ocean for OA
factors. MSA-OA is primarily associated with the Iceland Basin, with rapid production following coccolithophore blooms (lag of 1-2 days), while diatoms contribute to a slower formation process (lag of 9 days), reflecting distinct oceanic biological processes. In contrast, PMOA is sourced from more variable ecoregions, including the Southern Celtic Sea, West European Basin, and Newfoundland Basin, with additional contributions from chlorophytes and cyanobacteria at more southerly latitudes. Overall, these findings emphasise the need for
further long-term investigation to fully account for phytoplankton's taxa variability influence on aerosol composition and their broader impacts on aerosol-climate interactions.

**Keywords:** Submicron Marine Aerosols, Secondary Organic Aerosols, HR-ToF-AMS, PMF, phytoplankton




**1 Introduction**

The marine environment plays a critical role in regulating climate through sea spray and gas-phase emissions

from the oceans, via direct and indirect solar radiation effects and cloud formation governed by ocean biology, sea spray physicochemical properties and secondary reactions (Cochran et al., 2017). However, aerosols in the atmospheric marine boundary layer (MBL) remain a significant source of uncertainty in radiative forcing estimates (Rosenfeld et al., 2019; Wang et al., 2020) primarily due to limited knowledge about aerosol mass, chemical composition, and particle number distributions (Carslaw et al., 2017). Along with this, the origin of

marine organic aerosols (OA), specifically whether formed by primary or secondary processes, requires further investigation.

Primary Marine Organic Aerosols (PMOA) consist of sea spray aerosols, produced by bursting bubbles, film, jet, and spume drops (Ovadnevaite et al. 2014; Veron 2015; Villermaux et al. 2022) that carry sea salt particles enriched in biogenic organic aerosols (O'Dowd *et al*. 2004; Facchini *et al*. 2008). The majority (80 %) of fine

carbonaceous particles in the clean N.E Atlantic marine atmosphere has been shown to directly originate from phytoplankton activity as reported with dual carbon isotopes analysis (Ceburnis et al. 2011). The phytoplankton-OA link is particularly well-established, yet the topic is still highly debated as no clear full-picture consensus has been reached owing to widely changing temporal and geographic fluctuations (Lawler et al., 2024; Lewis et al., 2021; Seidel et al., 2022). In addition, the specifics of how phytoplankton control OA chemical composition

(Behrenfeld et al., 2019; Facchini et al., 2008; O'Dowd et al., 2015), numbers flux (Markuszewski et al., 2024; Sellegri et al., 2023), size (Croft et al., 2021; O'Dowd et al., 2004; Saliba et al., 2019), lifespan and surface tension (Lee et al., 2020; Ovadnevaite et al., 2017; Sellegri et al., 2021) are all the focus of intense ongoing investigations.

In a warming world, following a high-emissions scenario (RCP8.5) trajectory, climate change is projected to

drastically alter the geographic and seasonal variability of phytoplankton blooms in the N.E Atlantic (Asch et al., 2019). Furthermore, long term trends already show that the N.E Atlantic has experienced major changes in phytoplankton functional diversity over the last 60 years (i.e. -5% dinoflagellates decade[-1] whereas diatoms increased by 0.1% decade[-1]) due to rapid warming and various environmental transformations attributable to climate change (Bedford et al., 2020; Holland et al., 2023; Mutshinda et al., 2024). All of this strongly supports

the pressing needs for further investigations on phytoplankton-aerosol interactions as environmental stressors will result in significant non-linear effects and tipping points (Ban et al., 2022; Wolf et al., 2024) .

In contrast to PMOA, marine Secondary Organics Aerosols (SOA) in the remote MBL arise from new particle formation (NPF) and are governed by subtle chemical mechanisms. These include gas-to-particle conversion (Peltola et al., 2022; Zheng et al., 2021), oxidation of volatile organic compounds and consequent volatility

reduction that leads to condensation (Kroll *et al*. 2018; Hallquist *et al*. 2009), ion-induced nucleation of biogenic particles (Kirkby et al., 2016) and fission of organic biogels (Karl et al., 2013). SOA formation occurs through various processes such as homogeneous, heterogeneous and multiple phase reactions (Marais et al., 2016; McNeill, 2015) as well as photochemical reactions (Brüggemann *et al*. 2018). While various SOA molecular classes have been identified, the complexity of SOA, which consist of thousands of multifunctional compounds

(Goldstein and Galbally, 2007) including high molecular weight species and oligomers from diverse sources underscores the pressing need for continued exploration. All of this can now be partly described thanks to



continuous widespread progresses in aerosol mass spectrometry (DeCarlo *et al*. 2006; Laskin, and Nizkorodov 2012). The present study focuses on source apportionment, aiming to delineate the sources of marine OA, particularly distinguishing between SOA and PMOA sources.

Both SOA and PMOA  serve as cloud condensation nuclei (CCN) (Mayer et al., 2020), impacting cloud albedo and lifetime, leading to uncertainties in global chemistry-climate models (Bellouin et al., 2020). Radiative transfer implications from such interactions range from -2.65 to -0.07 Wm$^{-2}$, contrasting with $CO_2$ radiative forcing estimate of 1.83±0.18 Wm$^{-2}$ (Etminan et al., 2016). In pristine environments, SOA nucleation events significantly shape CCN concentrations, altering cloud radiative forcing (Liu and Matsui, 2022) but so does the

presence of primary sea spray (Fossum et al., 2018). Complementing this, previous literature shows that phytoplankton activity is related to emissions of organic and sulphate particle CCN precursors (O'Dowd et al., 2015; Sanchez et al., 2021). There is an ongoing debate over the respective impacts from primary sea spray (Ovadnevaite et al. 2011; King et al. 2012; Schwier et al. 2015; Xu et al 2021) and secondary aerosols (Mayer et al., 2020; Quinn et al., 2017) on cloud formation in pristine environments. This study, thus, aims at identifying

aerosol sources and quantifying elemental ratios which can serve as a proxy for ulterior parametrisations (e.g. as done in Han *et al*. 2022; Li *et al*. 2023)..

Finally, the effects from ocean biology are currently unaccounted for in climate models (Sellegri et al., 2021), therefore, this study seeks to relate different aerosol sources and geographical regions to phytoplankton taxonomic group simulations (Rousseaux *et al*. 2013). This multi-faceted approach allows to place the local

measurements at Mace Head into the broader context of ocean-atmosphere interactions and explore the potential influences of marine ecosystems on atmospheric aerosol loading over the North Atlantic region.

## 2. Materials and Methods
### 2.1 Site Description
Mace Head (MHD) atmospheric research station is located on the west coast of Ireland (53.33°N, 9.90°W) on a

peninsula exposed to open ocean air masses. These air masses, originating from a nominal clean sector (between 190° and 300°; Grigas *et al*. 2017) are predominantly steered by westerlies  ushered by the polar jet's low-pressure systems. Importantly, open ocean air mases are mostly devoid of anthropogenic influences, with over 60% of air masses arriving at MHD classified as pristine marine (Grigas et al., 2017; Sanchez et al., 2022). However, the remaining 40% of all the other air masses from the 360 sector exhibit varying degrees of

anthropogenic influences, particularly during or just after periods of continental outflow under high-pressure regimes (Jennings *et al*. 2003).

This study focuses on August 2015, a summertime period characterized by heightened biological activity (Behrenfeld et al., 2019) and predominant pristine marine conditions. This specific year is also marked by the onset of the *cold blob*, with the subpolar gyre region (The North Atlantic waters south of Greenland) reaching

around 2°C lower than previous long-term average possibly owing to the slowing down of the Atlantic Meridional circulation and Greenland Ice melt  (Rahmstorf et al., 2015; Sanders et al., 2022). As such these specific conditions could serve as an indication for future measurements of aerosols-phytoplankton interactions during future cold blob phenomena.





**2.2 In-situ measurements**

Ambient submicron non-refractory aerosol major species were monitored using an Aerodyne high-resolution time-of-flight aerosol mass spectrometer (HR-ToF-AMS) equipped with a standard tungsten vaporizer operated at 650°C. The instrument working principles have been extensively described in the literature (Canagaratna et al., 2007; DeCarlo et al., 2006). The HR-ToF-AMS used a 5 min time resolution scan on the single-reflection highly sensitive V mode configuration (mass resolution up to 3000 m/Δm) while detection limits were estimated

based on the approach described by Drewnick *et al* (2009). Ionisation efficiency (IE), particle velocity and inlet flow were determined following standard methods while applying standard RIEs (Nault et al., 2023; Xu et al., 2018) . The particle transmission and detection efficiency expressed as the collection efficiency (CE; Huffman *et al*. 2005) was corrected for detection losses due to particle bounce and lens efficiency by applying the composition-dependent collection efficiency (Middlebrook *et al*. 2012)..

The AMS data were analysed using SQUIRREL (SeQUential Igor data RetRiEvaL) v1.65B and PIKA (Peak Integration by Key Analysis) v1.25B software packages. Sea salt was estimated based on a scaling factor of 51 of the common sea salt ion $NaCl^+$ (m/z 57.96) (Ovadnevaite et al. 2012) while MSA was quantified by upscaling the $CH_3SO_2$ (m/z 79) ion (Ovadnevaite *et al*., 2014). Interferences from MSA on $SO_4$ and OA were accounted for as follows:

$$SO_{4\ corrected} = SO_4 - \frac{CH_3SO_2 * 12.48}{RIE_{SO_4}} \quad (1)$$

$$OA_{corrected} = OA - \frac{CH_3SO_2 * 15.86}{RIE_{Orgs}} \quad (2)$$

An improved Ambient (I-A) method was adopted for the mass spectra elemental ratio analysis of O:C, H:C, N:C, S:C, and the OM:OC (organic matter to organic carbon) ratio (Canagaratna et al., 2015). High-resolution analysis was performed on each m/z in the mass range 12–130 m/z with ion fitting applied to difference between

open and closed spectra. Based on their elemental composition (C, O, H, N, S), ions were then grouped into chemical families: $C_x$, $C_xH_y$, $C_xH_yO_z$ (z = 1), $C_xH_yO_z$ (z > 1), $C_xH_yN_w$ (w = 1), $C_xH_yN_w$ (w >1), $C_xS_j$, $H_yO_z$, $N_wH_y$, $N_wO_z$, $S_jO_z$, and $C_xS_i$ where the indices x, y, z, w, j represent the number of C, H, O, N, S atoms, respectively.

The concentration of equivalent black carbon (eBC) was measured by a multi-angle absorption photometer (MAAP, Thermo Fisher Scientific model 5012). The MAAP operated at a flow rate of 10 L min$^{-1}$ and a 5 min

time resolution. The transmittance and reflectance of eBC-containing particles were measured by the MAAP at two different angles to derive optical absorbance as detailed in (Xu et al., 2020).

Carbon monoxide (CO) measurements were also conducted using a model RGA-3 CO analyzer (Trace Analytical, Inc., CA, USA), which operates on the principle of hot mercuric oxide reduction gas chromatography (Derwent et al. 1994).

Ozone (O₃) was measured with an UV O₃ spectrometer (Model 8810, Monitor Labs San Diego, CA), the raw voltage output was converted to concentration values based on Automatic Urban/Rural Network (AURN) calibration audits (Derwent *et al*. 2018). Finally, meteorological data were continuously recorded at the station (including rainfall, solar radiation, wind speed, wind direction, temperature, relative humidity and pressure)



using standard meteorological instruments and retrieved using the *worldMet* R package (station ID: 039630-

99999) from the NOAA ISD website (https://www.ncdc.noaa.gov/isd).

### 2.3 Source apportionment

The HR organic mass spectra was deconvolved using the Positive Matrix Factorization (PMF; Paatero and
Tapper 1994; Paatero 1999) source-receptor model to investigate the various source contributions to OA. A
major advantage of using HR data over unit mass resolution is the distinct differentiation of multiple ions

sharing the same nominal mass, thereby allowing for a more exact characterisation of the temporal fluctuations
of different ion families (e.g., $C_xH_y^+$, $C_xH_yO_z^+$). The information richness in the HR-ToF-AMS datasets, as a
result of the improved chemical resolution, is advantageous for restricting the PMF solutions, minimizing
rotational ambiguity and results in more reliable solutions and a larger number of interpretable OA factors.

The IGOR PRO Source Finder (SoFi v6.8.1) toolkit (Canonaco *et al*. 2013) was used to run the PMF algorithm.

Solutions were assessed across 2 to 12 factors using the unconstrained factors rotational Fpeak tool. Factors were
explored for Fpeaks (rotations) between −1 and 1 (0.1 steps). A final solution consisting of 4 factors was retained
as the optimal solution based on several considerations. These include its Q/Qexp ratio value (1.38),which is
tested for a range of FPEAKS and scaled residuals distribution (as recommended by Zhang *et al*. 2011). The
solutions were also investigated in regard to key diagnostic plots, diurnal profiles, correlations with meaningful

external tracer time series and reference mass spectra (Canonaco et al. 2021) extracted from the aerosol mass
spectrometer database (Ulbrich et al., 2009).

### 2.4 Air masses trajectory analysis

Air masses back trajectories analysis was performed using the Hybrid Single Particle Lagrangian Integrated
Trajectory (HYSPLIT) (Stein et al., 2015). Meteorological data were accessed from the Global Data

Assimilation System (GDAS) archived by NOAA Air Resources Laboratory. HYSPLIT was used to calculate 72
hours back trajectories every 3 hours with starting height set to 100m above ground level. To investigate
potential source regions leading to total particle mass concentrations from each resolved source, the back
trajectories were gridded to 1° × 1° grid cells and linked to particle concentrations using trajectory source
contribution functions. While common source contribution functions assume that trajectories centrelines are

accurate, we focused instead on the Simplified Quantitative Transport Bias Analysis (STQBA) method which
considers plumes transport bias along air mass trajectories as a more robust approach.

The Boundary layer height (BLH) was determined from the fifth generation ECMWF (European Centre for
Medium-Range Weather Forecasts) atmospheric reanalysis (ERA5) dataset based on the bulk Richardson
number (Guo et al., 2021) by mapping HYSPLIT trajectories footprint along the gridded BLH data. This was

used to find the fraction of time spent over the Ocean, within the Marine Boundary Layer (MBL; altitude
<BLH), in the Marine Free Troposphere (MFT; altitude >BLH) and over land in the planetary boundary layer
(PBL; altitude < BLH). The R package *rnaturalearth* was also used to obtain a high-resolution land mask for
Ireland allowing for identification of purely marine air masses (no advection over land for at least 3 days prior to
being sampled at MHD) and aided in delineating lands from oceans.

Finally, NASA Ocean Biogeochemical Model (NOBM) taxonomic group simulations (Rousseaux *et al*., 2013)
for *coccolithophores*, *diatoms*, *chlorophytes* and *cyanobacteria* were used to visualise phytoplankton geographic
repartition estimates as well as for lags calculations with OA similarly to O'Dowd *et al* (2015).



**2.5 Transfer entropy analysis**

The R package *RTransferEntropy* (Behrendt et al., 2019) was used to quantify the information flow between
time series using the transfer entropy (TE) as previously done on recent SOA studies (Long et al., 2023; Sinha et
al., 2024). Transfer entropy (TE) is a prediction model that quantifies the directional influence between two time
series X and Y by determining how the past values of one series predict the future behaviour of the other
(Schreiber, 2000). TE is calculated using Rényi entropy, a generalisation of Shannon entropy that offers
enhanced robustness in the presence of tails effects. To account for spurious information transfer, the transfer
entropy is also estimated from a shuffled version of the time series. This shuffled estimate called *effective
transfer entropy* (eTE) is used to correct for sampling bias, ensuring the validity of the results. Statistical
significance is assessed with a bootstrapped Markov chain, where a p-value of less than 0.05 indicates a
significant information transfer between X and Y. The reader is referred back to Behrendt et al. (2019) for more
details.

**3. Results**

**3.1 Submicron aerosol chemical composition overview**

The mass concentration time series of organic aerosols (OA), methane sulphonic acid (MSA), sulfate ($SO_4^{2-}$),
nitrate ($NO_3^-$), ammonium ($NH_4^+$) and sea salt measured by the HR-ToF-AMS as well as eBC from MAAP
measurements are shown on Figure 1. The average chemical composition was dominated by OA (32%),
followed by $SO_4^{2-}$ (31%), sea salt (20%), MSA (7%), $NH_4^+$ (6%), $NO_3^-$ (2%) and eBC (2%) (Figure 1).

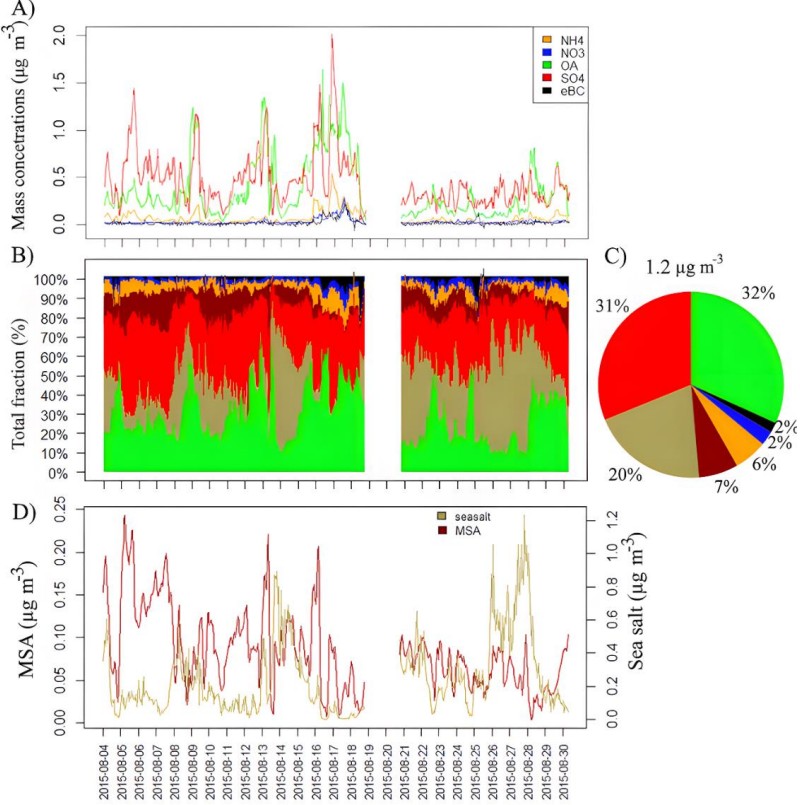



**Figure 1: A) OA, SO₄., NH₄, NO₃ and eBC mass concentrations time series – µg m⁻³ B) Relative contributions to total PM1 C) Pie plot of total contributions to total PM1 D) shows MSA and sea salt – µg m⁻³.**

The total average bulk submicron aerosol mass was 1.2 µg m$^{-3}$ over the entire measurement period. These high SO$_4^{2-}$ and OA relative contributions and overall low concentrations are common for coastal sites during summertime in the marine boundary layer as reported over the North & South Atlantic Ocean (Ovadnevaite *et al*., 2014; Huang *et al*., 2018) as well as in the Artic (Willis *et al*., 2017; Nielsen *et al*., 2019). MSA in particular showed mass concentrations values of 0.08± 0.04 µg m⁻³ in the range of those previously reported at Mace Head

(0.05±0.04) (Ovadnevaite *et al*. 2011) and more diverse locations such as the central Arctic (Dada et al., 2022) and the Atlantic Ocean from 53° N to 53° S where average mass concentrations of 0.04 ± 0.03 µg m⁻³ (Huang et al., 2018) were reported.

The low mass concentrations of NH$_4^+$, NO$_3$ and corresponding N:C ratio of 0.006 ± 0.002 (Figure S1), indicate a limited presence of amino acids (below detection limit) from usual sources such as the North Atlantic

oligotrophic gyre, ornithogenic emissions (i.e., birds), phytoplankton, bacteria, or in situ atmospheric processes (Schmale *et al*., 2013; Van Pinxteren *et al*., 2022).

Following eBC thresholds established for the North-East Atlantic (Grigas *et al*., 2017), pristine conditions (eBC levels below 0.015 µg m$^{-3}$) were observed during 60.4% of the measurement period. Clean conditions (eBC levels between 0.015 and 0.05 µg m$^{-3}$) prevailed 30.5% of the time, and moderately polluted conditions (eBC

levels between 0.05 and 0.3 µg m$^{-3}$) occurred for 9.1% of the time with a significant pollution event spanning from August 17$^{th}$ to 19$^{th}$ 2015 onwards.

Likewise, CO mixing ratios were below 100 ppb for over 70% of the time, similarly to other pristine sites (Zhao et al., 2022). Winds advected through the clean sector (190-300°) for over 78% of the time. Finally mean wind speed was 6.6 ± 3.1 ms⁻¹ only being below the whitecap threshold of 4 m s⁻¹ (O'Dowd *et al*., 2014)  for 23% of

the time hinting at strong  sea spray influences.

SOA influences were also consequent as revealed with the average OM/OC (organic matter to organic carbon ratio) value of 2.10 ± 0.14 (Figure S1), aligning with the value of 1.9 previously reported for clean aged marine polar air masses at MHD (Ovadnevaite et al. 2014). Additionally, AMS derived OM/OC values in the high Arctic (Nielsen et al., 2019) also fall within the range of 1.96 to 2.42 for PMOA (primary marine organic

aerosols) and MO-OOA (more oxidised organic aerosols) respectively, here median OM/OC value was 2.11 with minimum and maximum OM/OC values of 1.71 and 2.42 respectively. This indicates the presence of both PMOA (i.e., saturated hydrocarbons, unsaturated hydrocarbons and cycloalkanes) as well oxygenated SOA formed with photochemical processing during long range transport (Aiken et al., 2008; Simon et al., 2011).

To get a better sense of the aerosol sources, the respective contributions of the marine boundary layer (MBL),

marine free troposphere (MFT) and planetary boundary layer (PBL) are shown in Figure S2. Overall, the measurement period was dominated by marine boundary layer influences (91% of the time), with minimal marine free troposphere influences (8%) and extremely low land-influences from the planetary boundary layer (1%) further hinting at pristine marine conditions.



### 3.2 Source apportionment

To accurately classify and categorize the diverse sources of OA that are present at Mace Head, source
apportionment was performed utilizing the Positive Matrix Factorization (PMF) method on the organic mass
spectrum, which ranged from m/z 12 to m/z 130. The resulting chosen four-factor solution, as depicted in Figure
2 yielded a Q/Qexp ratio of 1.38 and accounted for up to 90% of the total measured organic aerosol mass.
Solutions with a higher number of factors introduced splitting and did not show additional emergent
interpretable sources (Figure S3, Text S1).

The following four factors, namely Methane Sulphonic Acid, More Oxidised Organics, Primary Marine
Organics and Peat, were determined as the optimal representation of the marine aerosol at Mace Head:

Methane Sulphonic Acid Organic Aerosols (MSA-OA): Representing approximately 17.2% of the total OA
mass, MSA displayed a distinct m/z fragment at m/z 78.98 ($CH_3SO_2^+$), accounting for 36.3% of its total mass
spectra signal intensity. The identification of specific methane sulphonic acid tracer ions further substantiated its
origin. More details on all factors are provided below in sections 3.2.1-3.2.4.

More Oxidised Organic Aerosols (MO-OOA): Making up about 31.8% of the total elucidated PMF solutions,
this factor exhibited prominent m/z fragments at m/z 27.99 and m/z 43.99 and showed significant correlations
with reference mass spectra for MO-OOA (R = 0.97) (Hu et al., 2015) and LO-OOA (R = 0.76) (Mohr et al.,
2012). O:C and H:C ratios were also in line with expected values for marine MO-OOA (Figure S4), correlations
with externals tracers and m/z ratios values further confirmed the MO-OOA desgination.

Primary Marine Organic Aerosols (PMOA): Comprising roughly 42.2% of the total resolved PMF solutions.
This factor exhibited m/z fragments similar to MO-OOA (Schmale et al., 2013), but with higher contributions
from aliphatics ($C_xH_y$) such as alkyls (dominant in sea spray during phytoplankton blooms; Cavalli et al. 2004),
alkenes (i.e. phenols or humic materials; Bahadur et al. 2010) and functional derivatives such as alcohols
($C_xH_yO_z$, where z=1) as established in earlier studies (Ovadnevaite et al. 2011; Crippa et al. 2013).

Peat-OA: accounting for approximately 8.8% of the total PMF solutions, was characterised by saturated
hydrocarbons ($C_xH_{2y+1}$), unsaturated hydrocarbons ($C_xH_{2y-1}$) and cycloalkanes ($C_xH_{2y}$) ion series. This factor
was identified as Peat-OA owing to its good correlation (R=0.75) with the Peat-OA reference mass spectra (Lin
et al., 2017). Additionally, its mass spectrum was marked by cellulose ($C_2H_4O_2^+$) at m/z 60 and by the
dominance of $C_3H_7^+$ rather than $C_2H_3O^+$ at m/z 43 which facilitated the distinction of peat emissions over wood
or smoky coal emissions.





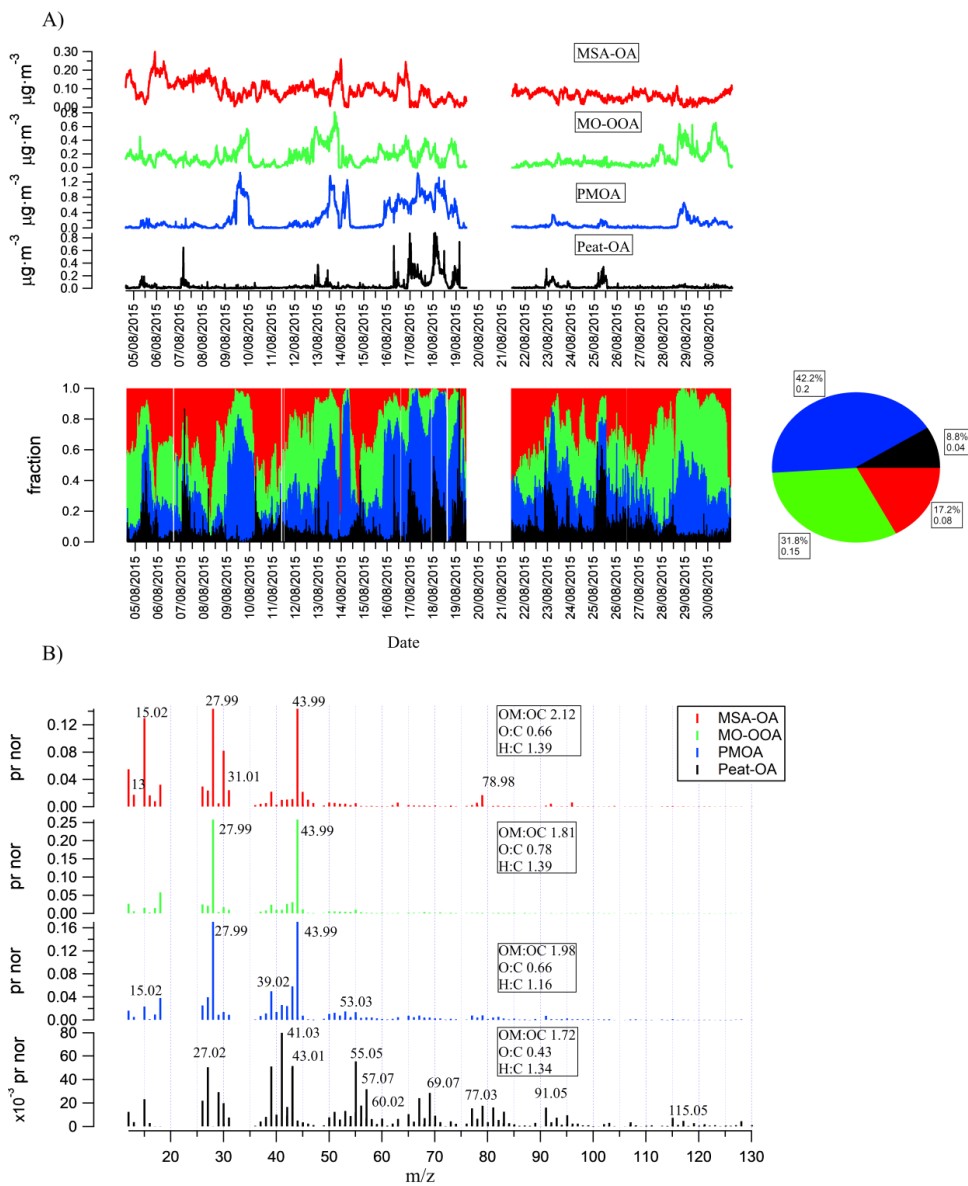

**Figure 2. A) Factors time series (MSA-OA in red, MO-OOA in green, PMOA in blue, Peat-OA in black) and associated relative contribution time series and pie chart showing fractions and respective mass concentrations (µg m⁻³) for the whole period B) Factors mass spectra (MSA-OA in red, MO-OOA in green, PMOA in blue, Peat-OA in black) and associated improved ambient OM:OC, O:C and H:C ratios (Canagaratna et al., 2015).**



### 3.2.1 More Oxidised Oxygenated Organic Aerosols (MO-OOA)

MO-OOA main contributing ions are associated with oxygenated compounds belonging to the COOH functional group (Figure S5), reflecting pronounced fragmentation of mono- and dicarboxylic acids into fragments with multiple oxygen atoms (Duplissy et al., 2011). Specifically, $C_xH_yO_z$ ($z > 1$) ions family accounts for 63.1% of the total mass spectra intensity, followed by $C_xH_yO_z$ ($z = 1$) ions family (m/z 27.99, m/z 43.02, m/z 42.01…) contributing 13.7% to MO-OOA, adding up to a total contribution of 76.8%. Additionally, $CO^+$ and $CO_2^+$ each

accounts for 25% of MO-OOA intensity which is typical for remote ocean carboxylic acids (Dominutti et al., 2022).

In contrast, $C_xH_y$ (aliphatics) ions family (m/z 13.00, m/z 15.02, m/z 16.03…) contributes only 13.7% to MO-OOA total mass spectra intensity. Nitrogen-containing ions fragments constituted a very low portion of the signal (0.8%), similarly to previous remote ocean measurements (Ovadnevaite et al. 2011). The weak contribution from

$C_2H_3O^+$ (3.2%) which has been reported to be predominantly due to non-acid oxygenates (Ng *et al*., 2011a) suggests a considerable prevalence of aging/oxidation during transport over the North-East Atlantic Ocean. This is also further confirmed by the low m/z 43:44 ratio of 0.12 hinting to MO-OOA rather than less oxidised species (Ng et al., 2011). This factor respective O:C ratio and H:C ratio of 0.78 and 1.17 further agree with MO-OOA reported at other similar locations (Figure S4). MO-OOA also has a strong contribution from $CO_2^+$ (25.7%) which

is assumed to originate mainly from acids or acid-derived compounds (Duplissy et al., 2011; Ng et al., 2011) that are known to be mostly water-soluble (Decesari et al., 2007) such as organic acids (e.g., mesotartaric acid, meso-erythritol, tartaric acid, oxalic acid) formed from oligomerization of small α-dicarbonyls (e.g., glyoxal) (Cui et al., 2022).

MO-OOA formation through ozonolysis is postulated based on a robust hourly averaged correlation (R=0.67) of

MO-OOA to $O_3$ across the entire observational period (Figure S6-B). Using the effective transfer entropy test (Behrendt et al., 2019) further reveals the non-linear dynamics between $O_3$ and MO-OOA, indicating $O_3$ as a significant reactant in the formation of MO-OOA from its precursors with a transfer entropy value of 0.014 (Figure 3) and an effective transfer entropy value of $0.0127 \pm 0.0009$ (p-value<0.05). In other words, there is a significant directional information flow between the two time series. Figure 3 also shows that MO-OOA is a mix

of local (Peat-OA) and regional marine influence (PMOA, but also MSA-OA to a lesser extent) all eventually concurring to MO-OOA formation, with ozone contributing 3x more information to MO-OOA than irradiance does. This aligns with studies showing $O_3$ to be a strong oxidation driver during summertime in the marine environment (Ovadnevaite *et al*., 2011), where unsaturated aliphatic chains (C=C double bonds) react with





ozone to form oxidised compounds (Decesari *et al*, 2011).

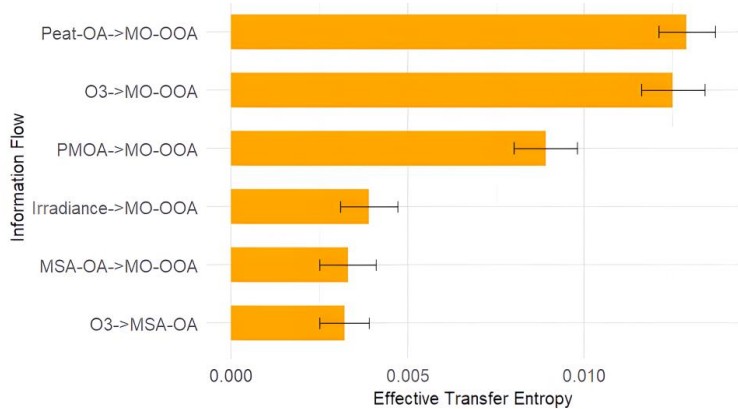


**Figure 3. Significant (p-value<0.05) effective transfer entropy flow values between PMF factors, ozone and irradiance.**

### 3.2.2 Methanesulphonic Acid Organic Aerosols (MSA-OA)

The mass profile of MSA-OA reveals that two oxygenated carbon families CHO (sum of $C_xH_yO_k$ and $C_xH_yO_w$

where $k = 1$ and $w > 1$) dominate 53.4% of the total mass spectra fraction followed by aliphatics (pure Hydrocarbon-like, $C_xH_y^+$) whose fraction accounts for 33.3% (Figure S5). MSA-OA is clearly identified owing to its substantial contribution from the $C_xS_y^+$ family (6%) over other sources, this is in line with the $C_xS_y^+$ contribution (7%) for MSA-OA also reported by Huang *et al*. (2018). The excellent correlation (R=0.82) between this factor and the $C_xS_y^+$ family (Figure S6-D) also further highlights the organosulphurs nature of

MSA-OA as opposed to other factors.

Similarly, to results reported by Schmale *et al*. (2013), the correlation coefficient with the AMS database MSA-OA laboratory reference spectrum (Figure S6-A) is rather moderate (R=0.55), although this factor spectra still allows for the precise identification of characteristic MSA ions at m/z 44.98 (CHS⁺), 47.00 ($CH_3S^+$), 64.97 ($HSO_2^+$), 77.98 ($CH_2SO_2^+$), 77.99 ($CH_3SO_2^+$), and 95.99 ($CH_4SO_3^+$). MSA-OA O:C and H:C ratios were 0.66

and 1.39 respectively, close to values (O:C: 0.54, H:C: 1.42) reported by Loh *et al*. (2022).

MSA-OA $C_xH_y$ family also features a typical $CH_3^+$ ion at m/z 15.02 that is absent from other factors. Similarly, the $C_xH_yO_w$ (w=1) family features the tracers ions $CH_2O^+$ (8.2%) and $CH_3O^+$ (2.4%) which are heat stress related markers (Faiola et al. 2015) attributed to methyl jasmonate (MeJA) and possibly acrylic acid (Van Alstyne and Houser, 2003) or other oxylipins stress enzymes (Aguilera et al., 2022; Koteska et al., 2022) which are known to

be emitted by kelp (Saha and Fink, 2022) or phytoplankton species (Koteska et al., 2022).

The $C_xS_y^+$ fragment family was dominated by CHS⁺ (25.9%), $CH_3SO_2^+$ (20.2%), $CH_2S^+$ (12.2%), $CH_4SO_3^+$ (7.5%), $CH_3SO^+$ (7.2%), $CH_2SO_2^+$, (6.9%), $CH_4SO_2^+$ (5.45), $CH_3S^+$ (6.4%), $C_2H_4SO_2$ (5.4%) and $CH_2SO^+$ (2.5%) which are common MSA ions found in the literature (Moschos et al., 2022). Overall, the $C_xH_y^+$ and $C_xS_y^+$ fragment ions families indicate a clear MSA fragmentation pattern with a characteristic high $CH_3^+$ relative intensity (13%)

typical for marine SOA in line with recent findings (Huang et al., 2018; Moschos et al., 2022).

Finally, MSA-OA correlated moderately (Figure S6-B) with particulate sulphate (R=0.51) which is expected since dimethyl sulphide, released by phytoplankton, can be oxidized to either form MSA or sulphur dioxide and then to sulphuric acid, leading to their partitioning into the particulate phase (Mungall *et al*, 2018).

### 3.2.3 Primary Marine Organic Aerosols (PMOA)

High-resolution mass spectrum of this factor reveals that two CHO oxygenated carbon families (sum of $C_xH_yO_k$ and $C_xH_yO_w$ where x = 1 and w > 1) dominate 61.5% of the total mass spectra followed by aliphatics (pure Hydrocarbon-like, $C_xH_y$) whose fraction accounts for 36.2% of the total mass spectra signal (Figure S5) aligning with previous findings reported by Ovadnevaite *et al* (2011). The $C_xH_yO_w$ (w=1) family features ions series (m/z 55.02, 69.03, 83.05, etc…) related to alkenyl groups, diunsaturates, cyclic alcohols, and ethers. Such functional

groups repartition is consistent with previous reports of water-insoluble organics being formed in sea spray (O'Dowd *et al*. 2004; Ovadnevaite *et al*. 2011). Additionally, this factor mass spectrum closely resembles (R=0.99) marine organic aerosols (MOA) mass spectra (Ovadnevaite *et al*. 2011) (Figure S6-A) and its O:C ratio of 0.66 and an H:C ratio of 1.16 respectively are close to literature O:C values for sea spray (Ovadnevaite *et al*. 2011; Flerus *et al*. 2012; Willoughby *et al* 2016) (Figure S4).

PMOA $C_xH_y$ mass spectra family is dominated by ion series $C_xH_{2y-3}$ (m/z 39.02, 53.03, 67.05 etc…) indicating dienes, alkynes, and cycloalkenes contributions, which is further confirmed by the presence of $C_xH_{2y-1}$ ions series (m/z 27.02, 41.04, 55.05 etc…) while the $C_xH_{2y+1}$ family (m/z 43.05, 57.07, 83.08 etc... y>2) indicative for anthropogenically influenced refined hydrocarbons is absent from this factor mass spectra. The marine biogenic origin of this factor is also indicated by the absence of alkanes ($C_xH_{y+2}$; m/z 16.03, 58.08, 72.09) which are

typical for continental air masses (Lewis et al., 2021) and by its lack of correlation with eBC (R=0.17) thereby excluding contribution from fossil hydrocarbons to PMOA. The $C_xH_y$ family is also marked by alkyls ($C_xH_{2y+1}$ ;m/z 15.02, 29.03, 37.00 etc…) which have been reported to be dominant in sea spray during phytoplankton blooms as a possible result of phosphate cycling (Cavalli, 2004; Meador et al., 2017).

Prior atmospheric measurements have shown that PMOA containing a large fraction of alkenes and oxygenated
functional groups (ie. alcohols, ethers, aldehydes, ketones) are dominated by insoluble organic colloids and aggregates (Facchini et al., 2008; Rinaldi et al., 2020) composed of microgels derived from phytoplankton extracellular metabolic extraction and adsorption organic pool rather than exopolymers produced from bacteria, with abacterial microgels aerosols being quite common and possibly accounting for 50-90% or phytoplankton derived organics (Bigg and Leck 2008; Bates et al. 2012; Liu et al. 2023). These bacterial exopolymers would
follow the makeup of ordinary bacterial cell fragments, which comprise approximately 55% nitrogen-containing organics and 10% carbohydrates (Schmale et al., 2013). The latter are accounted for by summing up pure carbohydrates (i.e.; glucose, saccharose, mannitol and glycogen) identified by typical fragments (Schmale et al., 2013; Schneider et al., 2011) at m/z 56.03 ($C_3H_4O^+$), m/z 60.02 ($C_2H_4O^+$), m/z 61.03 ($C_2H_5O^+$) and m/z 85.03 ($C_4H_5O+$) only amounting for about 1.3% of the total PMOA aerosols mass. Similarly, contributions from other
bacterial tracers such as glycogen; m/z 55.01 (1.4%), mannitol; m/z 56.02 (0.4%) and polysaccharide species; m/z 97.02 ($C_5H_5O_2^+$) and m/z 125.02 ($C_6H_5O_3^+$) (Glicker et al., 2022) tracer ions were also relatively poor (0.7%). All of this paired with below detection limit amino acids thus implicates that PMOA organic pool was largely shaped by abacterial processes. However, bacterial influence cannot be ruled out entirely as

carbohydrates might have been processed by enzymes or acidity during air masses transport and subsequent
aging (Zeppenfeld et al., 2023).

### 3.2.4 Peat Related Organic Aerosols (Peat-OA)

Although the measurement period is largely dominated by pristine ocean air masses, some residential heating
influence is still observed owing to local peat burning. Peat-OA (Figure S5) mass spectrum is largely dominated
by $C_xH_y^+$ ions (76.9%) such as alkyls-$C_xH_{2y+1}^+$ ($C_4H_7^+$ at m/z 55.05, $C_2H_5^+$ at m/z 29.03, $C_3H_7^+$ at m/z 43.05…),
alkenes-$C_xH_{2y-1}^+$ ($C_3H_5^+$ at m/z 41.03, $C_2H_3$ at m/z 27.02, $C_5H_7^+$ at m/z 67.05…) and cycloalkanes-$C_xH_{2y}^+$
(possibly $C_3H_3^+$ at m/z 39.02, $C_7H_7^+$ at m/z 91.05). This factor mass spectrum correlates well (R=0.86) with
previous measurements of Peat-OA in Galway city  (Lin et al., 2017) (Figure S6-A). More specifically, the
presence of aromatic ion series at m/z 77.03 ($C_6H_5^+$) and m/z 91.0.5 ($C_7H_7^+$) (Cubison et al., 2011) and the ratio
between m/z 55.05 ($C_4H_7^+$) and m/z 57.07 ($C_4H_9^+$) of 1.74 as well as the ratio between m/z 43.05 ($C_3H_7^+$) and
m/z 44.01 ($C_2H_3O^+$) of 1.03 all allow for the clear distinction of peat burning over other sources (Lin et al.,
2017). Peat-OA was freshly emitted as evidence by the pollution wind rose (Figure S7-E, S7- F) and concurrent
increase along with eBC (R=0.72) indicating that both were locally co-emitted within the planetary boundary
layer (PBL).

### 3.3 Elemental ratios -Van Krevelen diagram

The Van Krevelen (VK) diagram (Heald *et al*., 2010) provides valuable information on chemical evolution of
OA as demonstrated by subsequent marine aerosols studies (Ovadnevaite et al. 2014; Willis et al. 2017; Dada et
al. 2022). The VK plot of the PMF factors identified in this study superimposed with bulk OA O:C and H:C
values is depicted in Figure 4. Overall, the bulk OA slope of -1.18 and $\overline{Osc}$ values spanning over -1.8 to 0.8 in
the carbon oxidation state space indicates that higher levels of oxidation involving the generation of carboxylic
acids, and the subsequent breakdown of the carbon backbone are prevalent over the measurement period which
is consistent with MO-OOA functional groups (Heald *et al*.2010, Ng *et al*., 2011).  The O:C ratios for MO-OOA,
PMOA and MSA-OA all fall within the range of 0.64–1.15 reported for diverse OOA factors from previous
studies (Aiken *et al*., 2008; Jimenez *et al*., 2009). All PMF factors have H:C values lower than 2 which indicate
that they all contain unsaturated carbons capable of reacting with $O_3$ (Ovadnevaite et al. 2011). This is evidenced
by effective transfer entropy flow analysis (Behrendt et al., 2019)  between Peat-OA, PMOA, MO-OOA, MSA-
OA and $O_3$ values (Figure 4) which indicates that Peat-OA had the highest information transfer flow, making it
the most susceptible to ozonolysis, closely followed by PMOA and, to a lesser extent, MSA-OA. Both Peat-OA
(O:C=0.43, H:C=1.34) and PMOA (O:C=0.66, H:C=1.16) VK positions broadly fall in the area consistent with
lignin-like compounds (H/C = 0.6–1.5, O/C = 0.1–0.6; Park et al. 2022) which have been largely associated with
terrestrial origin OA (Jang et al., 2022) and found to be high in Arctic Ocean air masses as well  (Choi et al.,
2019) with authors reporting ~30% of the total assigned molecular formulae as marine lignin-like compounds.
These lignin-like compounds are also know to oxidise and form Humic-like molecules, characterized by polar
carbonyl (keto and carboxyl) functional groups alongside hydrophobic aliphatic chains (Cavalli, 2004) which
broadly agrees with MO-OOA functional groups.

MSA-OA (O:C=0.66, H:C=1.9) is then also examined by colouring the VK scatter plot (Figure S8) with the
MSA-OA/$SO_4$ ratio, a proxy for biological marine sources contributions from DMS (Chen *et al*., 2021) with



values ranging from 0.001 (ubiquitous anthropogenic influences) to 0.354 (significant contribution from biological marine sources) with an average value of 0.102 in line with pristine conditions (Huang *et al*., 2018).

Figure S8 shows that high MSA-to-sulphate ratio were consistent with VK regions for $C_2$-$C_{12}$ saturated diacids and inconsistent with $C_4$-$C_{12}$ carbohydrates (trehalose, erythritol, arabitol, mannitol, sucrose, galactose, glucose, fructose etc…) similarly to results reported for summertime Arctic aerosols (Wilis *et al*., 2017). However, as opposed to Artic aerosols, H:C ratios being higher, we report no association with VK areas for $C_4$ unsaturated diacids (e.g maleic and fumaric acid) nor with $C_{10}$ and $C_5$ keto-acids (levulinic and pinonic acid) which are

aqueous photochemistry tracers from isoprene and α-pinene oxidation (Kołodziejczyk et al., 2019; Rapf et al., 2017). This is in line with the absence of other isoprene tracers; $C_4H_5^+$ (0.5%) at m/z 53.03 and $C_5H_6O^+$ (0.2%) at m/z 82.04 (Hu et al., 2015; Robinson et al., 2011) and monoterpenes tracers (Boyd et al., 2015), namely $C_5H_7^+$ at m/z 67.05 (0.1%) and ($C_7H_7^+$) at 91.05 (0.1%) . This is also supported with the lack of covariance (Cov[X, Y] ≈ 0) between bulk $CO_2^+$, $CO^+$ and $C_2H_3O^+$ time series which also denotes the absence of non-acid

carbonyls (naCO) (Yazdani et al., 2022) which are known to be derived from isoprene and monoterpenes (Russell et al. 2011). The reasons behind the absence of isoprene and monoterpenes influence on OA in these findings are currently unclear although processes such as surface ocean consumption or unexplored oxidations pathways could be a possibility (Benavent et al., 2022).

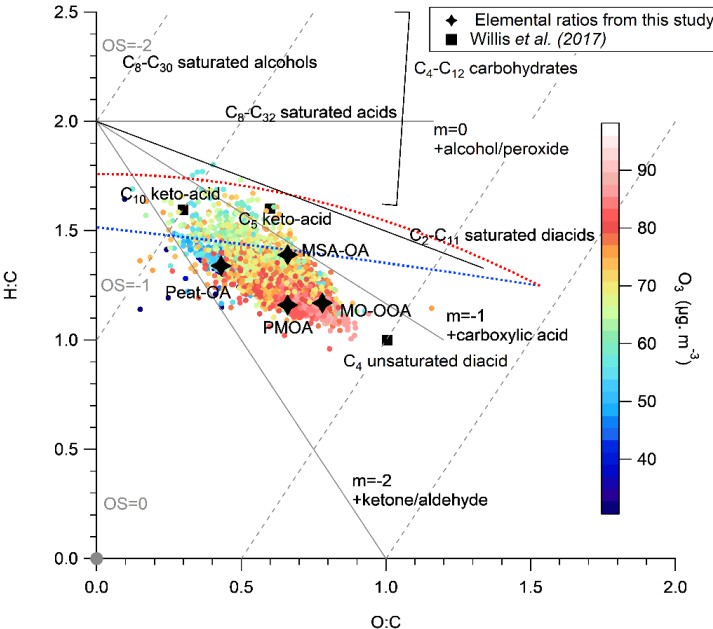

**Figure 4. Relationship between the ToF-AMS estimated hydrogen-to-carbon (H/C) and oxygen-to-carbon (O/C) ratios of organic species [Canagaratna *et al*., 2015] coloured by $O_3$ mixing ratio, all observations above ToF-AMS detection limits are shown for the entire period. Grey lines represent the ambient range of O/C and H/C observed by Ng *et al*. [2011] while dashed line represent the average carbon oxidation state (OSc ≈ 2 × O : C − H : C) (2011) superimposed on the Van Krevelen diagram (Ng *et al*. 2011, Kroll *et al*., 2011).**

**Elemental composition of C8–C30 saturated alcohols, C8–C32 saturated acids, C2 –C11 saturated diacids,**



**C4 unsaturated diacid (maleic and fumaric acid), C4 –C12 carbohydrates (e.g., trehalose, erythritol, arabitol, mannitol, sucrose, galactose, glucose, and fructose), and C5 and C10 ketoacids (levulinic and pinonic acid, respectively) are shown for reference (Willis *et al*., 2017).**

**3.4 Air masses and source apportionment**

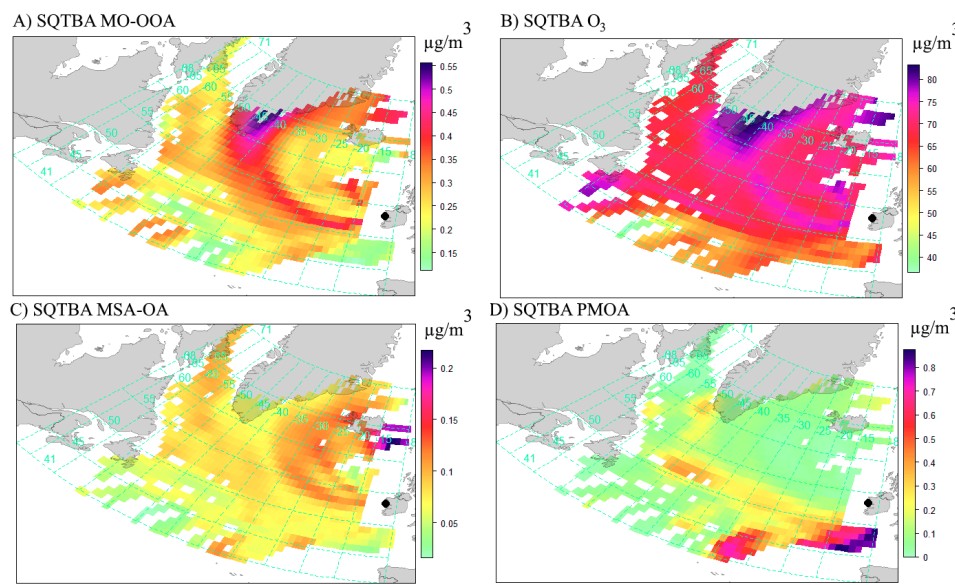


**Figure 5. Simplified Quantitative Transport Bias Analysis (SQTBA) -Gaussian Air masses dispersion for PMF sources (PMOA, MO-OOA, MSA-OA) and O$_3$.**

MO-OOA (Figure 5-A) strongest sources can be traced back northward along a cyclonic gradual crescent shape
spreading from Greenland Seas South of Cape Farewell (See Figure S9 for Ocean areas toponomy). This culminates further with air masses origins spanning over the East Greenland Current (Denmark Strait) upwards to the Iceland Sea south of Jan Mayen. MO-OOA is otherwise ubiquitous and shows contributions over the Newfoundland, Labrador and Iceland basins as well as other areas. O$_3$ (Figure 5-B) shared similar origin as MO-OOA further confirming its role in MO-OOA formation. Overall, we observe aged polar air masses eventually
flowing from Greenland to MHD. The sustained blockade and aging of air masses over Greenland is known and attributed to summertime high-pressure systems surrounding this region influenced by Arctic amplification (Pettersen et al., 2022; Preece et al., 2023) where Irminger current also acts as a hotspot for turbulent eddies and heat transport which might contribute to aerosol nucleation (Semper et al., 2022). Here the presence of a blocking anticyclone transition (Figure S10) leading to reduced cloud cover and warm air advection might
ultimately have contributed to an increase in aged SOA at the southern tip of Greenland possibly owing to its orography.





**Figure 6. Time averaged maps (0.67 x 1.25 deg) over 2015-Aug, Region 59W, 37N, 34E, 82N of dominant phytoplankton groups from NOBM Model data (Rousseaux *et al*. 2017; Buchard *et al*. 2017) and corresponding lagged crossed correlations for MSA-OA (red) and PMOA (blue) against A) Diatoms B) Coccolithophores, C) Chlorophytes and D) Cyanobacteria :blue and red shaded areas correspond to maximum significant crossed-correlations extracted from the autocorrelation function (ACF) 95% criteria.**





MSA-OA (Figure 5-C) main sources include the Iceland basin and more specifically the Iceland-Faroe Ridge. This is consistent with literature highlighting the diversity of eukaryotic phytoplankton in the Icelandic marine environment with the haptophyte coccolithophore *Emiliania huxleyi* being dominant during summertime (Cerfonteyn et al., 2023) owing to nutrients transport by the North Atlantic Current acceleration (Oziel et al., 2020) and findings (O'Dowd *et al*. 2015: Mansour *et al*. 2023) indicating concomitant MSA concentrations uptick during summertime. MSA-OA also spans along the East Greenland Current (Denmark Strait) where wind-driven coastal upwelling (Håvik and Våge, 2018) might result in increased DMS emissions (Edtbauer et al., 2020). Likewise, MSA-OA extend moderately over diverse regions such as the North-Western European Basin, the Newfoundland basin (where intense DMS fluxes have been reported; Bell *et al*. 2021) and the Labrador Sea.

PMOA (Figure 5-D) on the other hand strongly extends over the South of the Celtic Sea and West of the Bay of Biscay as well as West European basin waters and are otherwise diffused all over the North Atlantic Ocean with moderate intensity hotspots over the Newfoundland basin (Davis strait) possibly owing to an inflated subpolar gyre (Hátún et al., 2016).

Examination of NOBM model data (Figure 6) further reveals distinct MSA-OA and PMOA patterns. MSA-OA overlap with coccolithophores dominated ecoregions as well as diatoms ones. Similarly, diatoms also seem to contribute to PMOA sources, which is in line with recent results hypothesising that diatoms have a greater atmospheric significance than other eukaryotes due to their observed enrichment in PMOA (Alsante *et al*, 2021) whereas association with coccolithophores appears much weaker than for MSA-OA. Another distinction lies in PMOA overlapping with chlorophytes (*flagellates*, *Phaeocystis spp*) over the Western European basin. This geographic area hosts more than 512 chlorophyte species (Narayanaswamy et al., 2010) with recent reports of chlorophytes being one of the key contributors to marine productivity (Landwehr et al., 2021), further research is warranted to fully understand their role along other phytoplankton in this region during summertime. Likewise, cyanobacteria (combination of *Synechococcus*, *Prochlorococcus*, and nitrogen fixers such as *Trichodesmium*) might also contribute to PMOA more sparsely, especially at lower latitudes in the North Atlantic Ocean as previously reported (Baer et al., 2017).

Calculated lagged correlations (Figure 6) further pointed at MSA-OA being directly associated with coccolithophores (with a lag of -1 day) as well as diatoms (lag of -9 days), however no significant correlations were observed for either cyanobacteria or chlorophytes. As opposed to MSA-OA, the association between coccolithophores and PMOA doesn´t appear as meaningful (their autocorrelations are not statistically significantly different from zero). PMOA on the other hand are also associated with diatoms (lag of -5 days) and show unique associations with chlorophytes (lag of -10 days) as well as cyanobacteria (lag of -11 days).

Overall, association between OA enriched sea spray time series and phytoplankton groups remains controversial owing to a wide range of governing mechanisms as highlighted by previous studies using chl-α as a proxy to calculate cross correlation time lags over the North Atlantic which were found to vary between 8 days (Rinaldi et al., 2013) and 24 days (O'Dowd et al., 2015) depending on the period and length of measurements.

Late summer measurements (Mansour et al., 2020) show partially comparable lags to this study with a reported oceanic biological activity affecting aerosol properties within the order of 10-20 days. This delay roughly spans



over the full blooming to decaying phase transitions of an algal bloom (Lehahn et al., 2014) and is linked to the release of SSA-transferable organic matter in surface seawater by the interaction with marine viruses causing the demise of phytoplankton blooms (O'Dowd et al., 2015).

Here, by focusing on the lagged correlations between PMF factors and specific phytoplankton groups rather than bulk-OA and chl-α, this study´s findings indicate that PMOA is formed on such timescale from cyanobacteria and chlorophytes (lags of -11 and -10 days respectively) owing to atmospheric transport from the Western European basin whereas overwhelming diatoms influence results in a much shorter lag of -5 days. Additionally, MSA-OA is rapidly produced from coccolithophores blooms in 1-2 days. This reflects stressed, senescent, grazed, or virus-infected phytoplankton releasing high quantities of DMSP which rapidly oxidises to form MSA-OA (Mansour et al., 2020).

Finally, the interpretation of diatoms´ role on either MSA-OA or PMOA remains ambiguous as the -5 days lag with PMOA could hint at lipase activity concurring to self-aggregation and formation of free fatty acids during bloom potentially followed by a post-bloom (lag of -9 days with MSA-OA) with significantly different taxa or simply advection from remote eco-regions further closer to the Arctic which have been reported to host rich MSA producing diatoms communities as opposed to more southerly latitudes (Becagli et al., 2016).

## 4. Conclusions

This study leverages high-resolution online aerosol mass spectrometry source apportionment to investigate the chemical composition and sources of submicron organic aerosols representing marine environment during a summertime period marked by phytoplankton blooms. The results emphasise balanced mass contributions from POA (PMOA and Peat-OA) and SOA (MO-OOA and MSA-OA), each category accounting for approximately 50% of the total submicron organic aerosol mass, with distinct chemical compositions reflective of their varied origins.

One of this study´s key finding is that summertime polar air masses undergo significant ozonolysis over the remote ocean which happens to be largely driven by Greenland blocking air masses aging and anticyclonic conditions. Transfer entropy is introduced here to explain the dynamics of ozonolysis in this context revealing significant information transfer to MO-OOA during unsaturated aliphatic chains (C=C double bonds) breakdown of PMOA as well as MSA-OA to a lesser extent. However, this transfer entropy approach additionally shows that MO-OOA is also being formed locally from Peat-OA oxidation, as such, further studies will aim at exactly delineating open ocean versus locally produced MO-OOA.

Another essential takeaway is that OA not only reflect atmospheric chemistry and meteorology but may also serve as an indicator of marine ecosystems (i.e. MSA-OA enzymes stress makers and PMOA phytoplankton extracellular metabolic processes markers). Air masses trajectory analysis also show aerosols-phytoplankton ecoregions contributions with MSA-OA traced to the Iceland Basin and the Iceland-Faroe Ridge, with a rapid production burst (lag of 1-2 days) following coccolithophore blooms. Whereas relationship with diatoms show much longer lag (9 days) indicating fundamentally different oceanic biological processes. In contrast, PMOA is sourced from more diverse ecoregions (Southern Celtic Sea, West European Basin, and Newfoundland Basin), with additional chlorophytes and cyanobacteria influences from more southerly latitudes. All of this suggests that different phytoplankton taxa contributions to OA lead to specific m/z tracers and functional groups



repartition (i.e. sulphides as coccolithophores tracers, aliphatics as tracers for diatoms) though further investigation is needed to explore the biological processes and ecoregions specificities influencing this relationship. Overall, this study demonstrates the complex aerosol chemistry and diverse geographic origins

influencing POA and SOA formation in the Northeast Atlantic marine environment. Our findings emphasise the need for further long-term investigation to fully account for the various precursors and pathways contributing to OA, given their significant impacts on aerosol-climate interactions.

*Author contribution.* JO, COD and DC designed the research. DC, JO and KNF operated the instruments and

verified the raw data. EC and JO produced the postprocessed data and figures. JO, COD, LL, DC, WX and LC re-edited the manuscript. EC wrote the paper with support from all authors who commented on the paper.

*Competing interests.* The authors declare that they have no conflict of interest.

*Acknowledgments.* This work was supported by the EPA-Ireland and Department of the Environment, Climate and Communications and University of Galway College of Science and Engineering Postgraduate Fellowship n°127407. C. Lin acknowledges the support from the International Partnership Program of the Chinese Academy of Sciences (Grant No. 175GJHZ2022039FN). The authors would also like to acknowledge the support from the SFI FFP award (22/FFP-A/10611) and from the EPA-Ireland AC[3] network. Finally, we would also like to extend

our gratitude to Seraphine Hausser for looking into geopotential height anomalies and producing figure S10.

*Data availability.* Data available upon request.

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
