# Peer review of "Marine Organic Aerosols at Mace Head: Effects from Phytoplankton and Source Region Variability"

_EGUsphere, 2024_

## Author Comment (AC1)

**Review for 'Marine Organic Aerosols at Mace Head: Effects from Phytoplankton and Source Region Variability' by Chevassus et al., submitted for publication in Atmospheric Chemistry and Physics (ACP)**

Chevassus et al. present high-resolution time-of-flight aerosol mass spectrometer measurements at Mace Head to investigate the chemical composition of marine submicron aerosol particles and applied positive matrix factorization to identify sources contributing to coastal PM. Overall, the manuscript is well-written. Below, I provide some suggestions for further improving the manuscript.

**Quality of the figures:**

The resolution of the figures in the document is rather poor, making several texts difficult to read. While this applies to multiple examples, I will specifically address Fig. 6, where the x- and y-axis labels, for instance, appear quite small and blurry. To improve readability, either the font size should be increased or, more effectively, the resolution of the figures in the document should be enhanced.

Additionally, there are some details in the figures that seem to result from a lack of thorough editing. For example, in Fig. 6c under "Chlorophytes," remnants of previous text can still be seen, which were not entirely removed from the image. A similar issue is present under 'Cyanobacteria.'

Figures resolutions and consistency were improved throughout the paper.

Furthermore, in Fig. 1b, the total percentage of components exceeds 100%, which does not seem logical. In Fig. 1c, '2%' is partially cut-off. Another issue in Fig. 1d concerns the labeling of the x-axis with dates. The date labels are positioned directly on the ticks, making it unclear what specific time they represent. Ideally, I recommend shifting the date labels slightly to the right or left so that each date label appears between two ticks. This would clarify that the ticks represent midnight. As it stands, it is not entirely clear what time the ticks indicate.

These issues have been corrected; we express our gratitude to the reviewer for their attention to these details.

**Introduction:**

Overall, the introduction of the manuscript is well-written; however, it does not adequately prepare the reader for the subsequent content. For instance, it lacks any mention of the current knowledge on peat-derived organic aerosols, methane sulfonic acid organic aerosols, and oxidized oxygenated organic aerosols. Including a few sentences summarizing the current understanding of these aerosol groups would be beneficial.

On the other hand, the current manuscript includes a fairly detailed discussion of cloud condensation nuclei (CCN). However, since the manuscript does not provide further

measurements or establish a significant connection to CCN, I would recommend shortening this section.

Additionally, it would be helpful to briefly explain the principle of Positive Matrix Factorization (PMF) and highlight how it has been utilized in previous studies by other research groups. This would provide readers with better context for understanding its application in the manuscript.

The introduction was updated to include the background on oxidised aerosols, MSA and peat. The CCN section was shortened and more details on the PMF and references to previous HR-ToF-AMS PMF studies in the marine environment were added instead

"The present study focuses on source apportionment in a coastal environment, with the aim to separate primary OA (POA) and SOA into their respective sources. Marine SOA sources notably include methane sulphonic acid that is formed through the oxidation of dimethyl sulphide (Becagli et al., 2019; Hodshire et al., 2019; Mansour et al., 2024), and oxidised OA (i.e. carboxylic acids; Kawamura and Bikkina, 2016) that are complex mixtures resulting from unsaturated fatty acid oxidation found in very diverse locations (Crippa et al., 2013; Florou et al., 2024; Nøjgaard et al., 2022). On the other hand, POA sources not only include sea spray but also potential anthropogenic influences like local biomass burning (i.e. wood, peat or charcoal) or long-range continental transport (Lin et al., 2019; O'Dowd et al., 2014; Xu et al., 2020). The source apportionment was performed with the positive matrix factorisation (PMF) model which has been widely adopted for more than two decades now (Paatero, 1999; Paatero and Tapper, 1994) and successfully used on a wide range of different instruments; HR-Tof-AMS (e.g. Aiken et al., 2008), ToF-ACSM (e.g.. Fröhlich et al., 2015), PTR-MS (e.g. Slowik et al., 2010), EESI-ToF (e.g. Tong et al., 2022), offline filters (e.g. Maykut et al., 2003), SMPS (e.g. Nursanto et al., 2023) as well as other matrix-based measurements. Several previous remote ocean HR-ToF-AM PMF studies have been carried out in the Atlantic (Crippa et al., 2013; Huang et al., 2018), Arctic (Moschos et al., 2022; Nielsen et al., 2019; Nøjgaard et al., 2022), Mediterannean Sea (Florou et al., 2024; Mallet et al., 2019), Pacific Ocean (Loh et al., 2023, 2024) and Antarctica (Giordano et al., 2016; Paglione et al., 2024; Schmale et al., 2013) which facilitates cross-sites comparability."

**Further comments:**

L54: You mention that sea spray aerosol particles are enriched in biogenic organic aerosol. Could you please clarify relative to, or in comparison with, what this enrichment is observed?
Submicron Sea spray aerosol particles are notably enriched in organics compared to supermicron sea spray particles. However, the text has been revised for clarity.

"Primary Marine Organic Aerosols (PMOA) are part of sea spray aerosols, produced by wave breaking (Ovadnevaite et al. 2014; Veron 2015; Villermaux et al. 2022) and made of biogenic organic matter (O'Dowd et al. 2004; Facchini et al. 2008)."

L120: In section 2.2, you mention that submicron aerosol particles have been measured. Does 'submicron' strictly refer to particles smaller than 1 micrometer, or is there a more specific cutoff for this classification?

The HR-ToF-AMS size range is approximately 35 nm to >1 µm in vacuum aerodynamic diameter (DeCarlo et al., 2006). The clarification was added to the manuscript.

L176: Maybe "arrival height" might be more appropriate than 'starting height'?#

The text has been edited, thank you.

L235 I think it would be easier to read if this part were rephrased as: '....where 77% of the time exceeds the for the white cap threshold of 4 m s$^{-1}$ hinting at strong sea spray influences.'

Thank you for this suggestion, the text has been edited for clarity.

"wind speed was 6.6 ± 3.1 ms-1 exceeding the whitecap threshold of 4 m s-1 (O'Dowd et al., 2014) for 77% of the time, hinting at strong sea spray influences."

L243 As I'm not a meteorologist, this question might seem basic, but could you please clearly define how you distinguish between the marine boundary layer, marine free troposphere, and planetary boundary layer? As I understood it, the marine boundary layer is a subset of the planetary boundary layer, yet here you present them as two separate categories. Could you elaborate on this distinction?

Thank you for this question, you are right, the "planetary boundary layer" includes both the "marine boundary layer" and "boundary layer over lands" (which was referred to as "planetary boundary layer").

To make the text clearer, the manuscript has been edited throughout to refer explicitly to "boundary layer over land" rather than "planetary boundary layer" for air masses over lands. Adding up the MBL and the marine free troposphere (MFT) gives the entire troposphere vertical column over oceans, whereas adding up the "boundary layer over land" and free troposphere (FT) gives the entire troposphere vertical column over lands. The separation between the boundary layer and Free troposphere is obtained by comparing the HYSPLIT trajectories altitudes against the ERA5 retrieved boundary layer (If the HYSPLIT trajectory altitude is lower than the ERA5 boundary layer then the air mass is the boundary layer and vice-versa if the HYSPLIT trajectory altitude is higher than the ERA5 boundary layer then the air mass is in the Free troposphere)

L275 Does this fragment $C_2H_4O_2^+$ clearly identify the molecule 'cellulose', or does it refer to carbohydrates or polysaccharides in general? Could you elaborate on your conclusion regarding cellulose?

C$_2$H$_4$O$_2^+$ (m60) is a typical m/z cellulose fragment in biomass burning representative of levoglucosan (a pyrolysis byproduct of cellulose). The text has been edited to refer to levoglucosan rather than cellulose to avoid confusion.

L421 Please rephrase to 'carbohydrates and derivatives' This is necessary because arabitol and mannitol are not strictly carbohydrates, but rather 'sugar alcohols'.

The text was edited, thank you for correcting this.

Please check to what extent the paper by Paglione et al. (2024) can be used for comparison here. In my opinion, positive matrix factorization was also performed in that study. A thorough comparison or at least a mention in the paper seems appropriate.

Paglione, M., Beddows, D. C. S., Jones, A., Lachlan-Cope, T., Rinaldi, M., Decesari, S., Manarini, F., Russo, M., Mansour, K., Harrison, R. M., Mazzanti, A., Tagliavini, E., and Dall'Osto, M.: Simultaneous organic aerosol source apportionment at two Antarctic sites reveals large-scale and ecoregion-specific components, Atmospheric Chemistry and Physics, 24, 6305–6322, https://doi.org/10.5194/acp-24-6305-2024, 2024.

Thank you for your recommendation. We observe similar sources to the PMF results reported in Paglione et al. (2024), namely with sea spray (PMOA) and MSA. The authors present some PMF results with two PMOA factors (one with polyols, saccharides, aliphatics and another one with polyols and lactic acid owing to sugar fermentation), two SOA profiles (one with MSA and DMA and another one with TMA and MSA) as well as one last factor being a mix of POA and SOA. A reference to this paper was added in the PMOA as well as in the MSA-OA sections of the paper.

---

## Author Comment (AC2)

The study by Chevassus et al. on Marine Organic Aerosols at Mace Head effectively highlights the influence of phytoplankton and source variability on different organic aerosols, analyzed using AMS. The manuscript is well-written and presents valuable findings, making it deserving of publication in the journal after addressing a few revisions.

1. The legibility of the text in the figures requires improvement, particularly in Figures 5 and 6. Ensure that labels and the fonts are clearly readable. Additionally, the labeling across all figures should be harmonized for consistency. For example, compare the font sizes of the labels in Figures 1 and 2. Introduce adequate spacing between panels to clearly distinguish factors and profiles.

We express our gratitude to the reviewer for their attention to these details. Figures resolution and formatting were improved throughout the manuscript.

2. The abstract does not mention whether the expected aim of obtaining elemental ratios for parameterization was achieved. If successful, explicitly state the derived ratios in the abstract and clarify their relevance. For which specific parameterizations can these ratios now be utilized?

Thank you for this question. Parametrisations from elemental ratios are beyond the scope of this paper and will be the subject of future work, the introduction was rewritten to reduce the emphasis on this aspect. Elemental ratios were still successfully obtained for each factor and showed aliphatic and lignin-like compounds contributing to more oxidised organic aerosol formation. This information was added up in the abstract:

"Elemental ratios (O:C-H:C) were derived for each of these factors: PMOA (0.66-1.16), MO-OOA (0.78-1.39), MSA-OA (0.66-1.39) and Peat-OA (0.43-1.34). The specific O:C-H:C range for MO-OOA hints at aliphatic and lignin-like compounds contributing to more oxidised organic aerosol formation"